# Human Papillomavirus (HPV) Vaccine Uptake and the Willingness to Receive the HPV Vaccination among Female College Students in China: A Multicenter Study

**DOI:** 10.3390/vaccines8010031

**Published:** 2020-01-16

**Authors:** Dingyun You, Liyuan Han, Lian Li, Jingcen Hu, Gregory D. Zimet, Haridah Alias, Mahmoud Danaee, Le Cai, Fangfang Zeng, Li Ping Wong

**Affiliations:** 1Department of Epidemiology, School of Public Health, Kunming Medical University, Kunming 650500, China; caile@kmmu.edu.cn; 2Department of Epidemiology, Zhejiang Provincial Key Laboratory of Pathophysiology, School of Medicine, Ningbo University, Ningbo 315200, China; hanliyuan@nbu.edu.cn (L.H.); 1811101025@nbu.edu.cn (L.L.); 176000723@nbu.edu.cn (J.H.); 3Department of Pediatrics, School of Medicine, Indiana University, 410 W, 10th St., HS 1001, Indianapolis, IN 46202, USA; gzimet@iu.edu; 4Centre for Epidemiology and Evidence-Based Practice, Department of Social and Preventive Medicine, Faculty of Medicine, University of Malaya, Kuala Lumpur 50603, Malaysia; haridahalias@gmail.com (H.A.); mdanaee@um.edu.my (M.D.); 5Department of Epidemiology, School of Medicine, Jinan University, Guangzhou 510632, China; zengff@jnu.edu.cn

**Keywords:** HPV vaccination uptake, willingness to receive, female college student, China

## Abstract

Background: This study aimed to determine human papillomavirus (HPV) vaccine uptake and willingness to receive HPV vaccination among female college students, in China, and its associated factors. Methods: An online cross-sectional survey of female college students across the eastern, central, and western regions of China was undertaken between April and September 2019. Partial least squares structural equation modeling (PLS-SEM) was used to examine factors associated with the HPV vaccine uptake and willingness to receive the HPV vaccine. Results: Among the total 4220 students who participated in this study, 11.0% reported having been vaccinated against HPV. There are direct effects of indicators of higher socioeconomic status, older age (β = 0.084 and *p* = 0.006), and geographical region (residing in Eastern China, β = 0.033, and *p* = 0.024) on HPV vaccine uptake. Higher knowledge (β = 0.062 and *p* < 0.000) and perceived susceptibility (β = 0.043 and *p* = 0.002) were also predictors of HPV vaccine uptake. Of those who had not received the HPV vaccine, 53.5% expressed a willingness to do so. Likewise, social economic status indicators were associated with the willingness to receive the HPV vaccine. Total knowledge score (β = 0.138 and *p* < 0.001), both perceived susceptibility (β = 0.092 and *p* < 0.001) and perceived benefit (β = 0.088 and *p* < 0.001), and sexual experience (β = 0.041 and *p* = 0.007) had a positive and significant direct effect on the willingness to receive the HPV vaccine, while perceived barriers (β = −0.071 and *p* < 0.001) had a negative effect on the willingness to receive the HPV vaccine. Conclusions: Geographical region and socioeconomic disparities in the HPV vaccination uptake rate and willingness to receive the HPV vaccine provide valuable information for public health planning that aims to improve vaccination rates in underserved areas in China. The influence of knowledge and perceptions of HPV vaccination suggests the importance of communication for HPV immunization.

## 1. Introduction

Cervical cancer is the most common cancer of the female genital system, in China. It ranks as the eighth most frequent cancer among women in China and is the second most prevalent cancer in women aged between 15 and 44 years [1,2]. The National Central Cancer Registry of China reported an estimated 4.29 million new incident cases (12,000 per day) and 2.81 million deaths (7500 per day) from cervical cancer, in China, in 2015. This corresponds to an age-standardized incidence rate of 201.1 cases per 100,000 and an age-standardized mortality rate of 126.9 deaths per 100,000 [3]. Recently, increasing evidence has shown a significant rise in cervical cancer incidence among Chinese women aged below 35 years. Nearly a third of new cases are now found in women aged 35 and younger [4]. Although the cervical cytology test has been considered an effective method to prevent cancer, and although China has provided free cervical screenings in some rural areas for women aged 35 to 59 since 2009 [5], evidence has shown that the uptake of the cervical cytology test rates remains alarmingly low among women in China [6,7].

The introduction of the HPV vaccine in many countries has resulted in major decreases in the specific human papillomavirus (HPV) infection rates. More than 10 years after the licensure of the first HPV vaccines, 99 countries and territories worldwide have introduced HPV vaccination programs [8]. A recently published systematic review and meta-analysis that included data from 60 million individuals and up to eight years of post-vaccination follow-up showed compelling evidence of the substantial impact of HPV vaccination programs on HPV infections and the incidence of associated diseases [9]. This encouraging finding suggests promising future prospects of HPV vaccination reducing cervical cancer incidence and mortality. It is also anticipated that the HPV vaccine should benefit China, especially as the country is facing challenges regarding cervical screening. Most importantly, recent evidence from the first HPV vaccine trial in China showed that, among Chinese women, the quadrivalent human papillomavirus (4vHPV) vaccine demonstrated robust and sustained efficacy against HPV6/11/16/18-related persistent infections and high-grade cervical disease for up to 6.5 years of follow-up [10]. The vaccine was also well tolerated with no evidence of vaccine-related serious adverse events in the Chinese population [10]. A decade after the United States Food and Drug Administration licensure in 2006, the first HPV vaccine was approved in China in 2016. The GlaxoSmithKline’s bivalent vaccine, Cervarix®(GSK, Rixensart, Belgium), and Merck’s 4vHPV vaccine, Gardasil® (Merck & Co., Inc, New Jersay, USA), were approved for use among women in China, by the China Food and Drug Administration, in July 2016 and April 2017, respectively.

Several studies have examined willingness and intention to receive HPV vaccines among Chinese females prior to the availability of the HPV vaccine in the Chinese market [11,12]. Now that the HPV vaccine is available in the Chinese market, it is vitally important to provide sound evidence regarding the actual HPV vaccine uptake rate, particularly among young adult women. There are several reasons for this. First, young adult women are at the age at which it is best to receive the HPV vaccine because they had no access to the HPV vaccine when they were adolescents. Secondly, owing to the high prevalence of HPV infection [13,14], the high infection rate among young Chinese females [15] as compared with older adults, and a variety of biological and behavioral reasons [14], sexually active young female adults are at a particularly higher risk of acquiring sexually transmitted infections (STIs). It is well established that college-aged students represent an important catch-up population of young adult women for HPV vaccination. The college environment presents new opportunities for exposure to STIs. In China, and indeed across the globe, evidence is mounting that STIs are on the rise among college students and that college students are experiencing a high STI burden [16,17,18]. In addition, the prevalence of HPV infection among Chinese women peaks between the ages of 20 and 24 [13], when most are in their college or undergraduate years.

Currently, three years after the introduction of the HPV vaccine in China, information about vaccine uptake and its acceptance among young adult Chinese women, particularly those at college or university, is scarce. However, to the best of our knowledge, no study has reported real HPV vaccine uptake among young women in China since it became available in the Chinese market. Information about HPV vaccination rates is crucial to enable the Chinese government to plan or carry out effective strategies necessary to optimize uptake of the HPV vaccine.

The purpose of this study was to explore the uptake of HPV vaccination among young adult college women in China and its associated factors. The second objective was to identify factors associated with the willingness to receive the HPV vaccine among the unvaccinated female college students.

## 2. Materials and Methods

### 2.1. Study Participants and Survey Design

Using a cross-sectional design and convenience sampling method, young female college students were recruited from a total of 136 universities throughout eastern (78 universities), western (31 universities), and central (27 universities) regions of China. Participating universities were those that responded positively to an invitation to assist with data collection. A survey was conducted between April and September 2019. The inclusion criteria were all female students from the participating universities. The survey was administered online. The link to the survey questions was sent to administrators and lecturers from all invited universities for dissemination to registered female students. In an attempt to achieve comprehensive recipient coverage, the link to the survey was also sent to students’ social media groups or forums. Students who had received the survey link were also encouraged to forward the link to their peers.

### 2.2. Instruments

The questionnaire consisted of sections that assessed demographic characteristics, risk behaviors, knowledge, and health beliefs related to HPV infection and vaccination, HPV vaccine uptake, and willingness to receive the HPV vaccine.

#### 2.2.1. Demographics and Sexual Risk Profile

Personal details including age, ethnicity, study program, and year of study were collected. The participants were asked to indicate whether they had partaken in any sexual activity, including vaginal, oral, and anal sex. The participants who had done so were asked about safe sex practice and their number of sexual partners. Safe sex was defined as sexual intercourse with the use of a male condom.

#### 2.2.2. Knowledge of HPV and HPV Vaccination

The participants’ knowledge was assessed using a series of questions regarding HPV infection, the relationship of HPV infection with the development of cervical cancer and genital warts, and HPV vaccination (20 item scale). The response options were “true”, “false”, or “don’t know”. A correct response was given a score of 1 and an incorrect or “don’t know” response was scored 0. The possible total knowledge score ranged from 0 to 20, with higher scores representing higher levels of knowledge.

#### 2.2.3. Beliefs Surrounding HPV and HPV Vaccination

Health belief model (HBM) derived items were used to measure the participants’ beliefs about HPV vaccination [19,20,21]. The questions probed perceived susceptibility to HPV (three items), perceived severity of HPV infection (three items), perceived benefits of HPV vaccine (three items), and perceived barriers to getting a vaccination against HPV (two items). The response option was “agree” or “disagree”.

#### 2.2.4. Practice and Willingness to Receive HPV Vaccination

The participants were asked to indicate their HPV vaccination status. Those who had not received the HPV vaccination were asked about their willingness to receive the HPV vaccine in the near future. The response option was “yes” or “no”.

The items in the questionnaire were content validated by several panel experts to ensure the relevance and clarity of the questions. After minor amendments, the questionnaire was pilot tested on randomly sampled students who were not included in the main study.

### 2.3. Ethical Considerations

This study was approved by the Medical School of Ningbo University Review Board, China (reference number 20190506). The students were informed that their participation was voluntary, and consent was implied on completion of the questionnaire. All responses were collected and analyzed without identifiers.

### 2.4. Statistical Analysis

All statistical analyses were performed using the Statistical Package for the Social Sciences, version 20.0 (IBM Corp., Armonk, NY, USA). A p-value of less than 0.05 was considered statistically significant. The reliability of knowledge items was evaluated by assessing the internal consistency of the items representing the knowledge score. The 20 knowledge items in the study sample had a reliability (Kuder–Richardson 20) of 0.931. Partial least squares structural equation modeling (PLS-SEM) was used to quantify the contributing factors of HPV vaccine uptake and willingness to receive the HPV vaccination. All variables showing statistically significant associations with HPV vaccine uptake and willingness to receive the HPV vaccination in the univariate analyses were included in the PLS-SEM. A bootstrapping approach was used to evaluate the significance of associations in the proposed model. Bootstrapping includes the random resampling of the original dataset to generate new samples of the same size as the original dataset. This technique not only assesses the reliability of the dataset but also the statistical significance of the coefficients and the error of the estimated path coefficients [22]. All PLS-SEM analyses were performed using SmartPLS 3.2.8 software (SmartPLS GmbH, Bönningstedt, Germany).

## 3. Results

### 3.1. Characteristics of Participants

A total of 136 universities from eastern (*n* = 78), central (*n* = 27), and western (*n* = 31) regions of China responded to the invitation to assist in recruiting participants for this survey. A total of 4220 completed responses were received between April and September 2019. The majority of the responses were from western China (40.8%), followed by eastern (38.4%), and central China (20.8%). A summary of the characteristics of the respondents is provided in the first column of Table 1.

### 3.2. Knowledge about HPV and HPV Vaccination

Figure 1 shows the proportion of correct responses to knowledge items for vaccinated and unvaccinated participants. The proportion of correct responses for all the knowledge items was higher for vaccinated participants than for the unvaccinated participants. Among the items with a low proportion of correct responses were “there is a cure for HPV infection”, “HPV can cause oral cancer” and “most people with HPV do not experience any symptoms”. The mean total knowledge score of the vaccinated participants was 13.1 (SD ± 4.6) and it was significantly higher (t(4218) = 6.575, *p* < 0.001, and Cohen’s d = 0.33) than the mean total knowledge score of the participants who had not received the HPV vaccine (11.5 ± 5.1).

### 3.3. Health Beliefs Regarding HPV and the HPV Vaccination

Figure 2 shows the proportion of agreement with HBM-derived items for vaccinated and unvaccinated participants. For items in the domains of susceptibility and perceived benefit, participants who received the HPV vaccination had a higher proportion of agreement than those who had not received the vaccination. However, there was a slightly higher agreement among the unvaccinated participants in two of the three perceived severity items. For perceived barriers to HPV vaccination, participants who had not received the vaccination reported a higher agreement for the item, “fear of side effects of HPV vaccination”.

### 3.4. HPV Vaccination Uptake

A total of 463 participants (11.0%, 95% confidence interval (CI) 10.1 to 11.9) reported that they had been vaccinated against HPV, and 3757 participants (89.0%, 95% CI 88.1 to 89.9) reported that they did not have the vaccination. Univariate analyses indicated that HPV vaccine uptake was associated with older age, being in a higher year of study, place of birth in a city area, residing in eastern China, higher maternal and paternal education levels, a higher monthly disposable fund, perceived family economic status being rich or intermediate, having ever had sexual experience, a higher total knowledge score, a higher perception of susceptibility, and perceived effectiveness of the HPV vaccine (Table 1).

Figure 3 shows the factors affecting HPV vaccine uptake analyzed using PLS-SEM. The model showed that older age (β = 0.084 and *p* = 0.006), being born in a city area (β = 0.058 and *p* = 0.002), residing in eastern China (β = 0.033 and *p* = 0.024), a higher monthly disposable fund, a higher total knowledge score (β = 0.062 and *p* < 0.001), and a higher perceived susceptibility (β = 0.043 and *p* = 0.002) had a positive and significant direct association with HPV vaccine uptake. The adjusted R2 for vaccination status in this model was 0.04, which indicates that 4% of HPV vaccination uptake could be explained by this model.

### 3.5. Willingness to Receive the HPV Vaccination

Among the 3757 participants who had not received the HPV vaccination, 53.5% (*n* = 2010) responded “yes” and 46.5% (*n* = 1747) answered “no” to willingness to receive the HPV vaccine. As shown in Table 2, there were significant associations between most of the variables investigated and willingness to receive the HPV vaccine in the univariate analyses (Table 2).

Figure 4 shows the factors affecting willingness to receive the HPV vaccination studied using PLS-SEM. The model showed that demographic characteristics, namely a higher family economic status (β = 0.03 and *p* = 0.039), a higher monthly disposable fund (β = 0.045 and *p* = 0.007), higher maternal education (β = 0.038 and *p* = 0.004), attending a large scale university (β = 0.037 and *p* = 0.012), and a higher year of study (β = 0.094 and *p* < 0.001) had significant direct associations with willingness to receive the HPV vaccine. Total knowledge score had the highest beta coefficient (β = 0.138 and *p* < 0.001), indicating significant relevance of knowledge in predicting willingness to receive the HPV vaccination. For the HBM construct, both perceived susceptibility (β = 0.092 and *p* < 0.001) and perceived benefit (β = 0.088 and *p* < 0.001) had positive and significant direct associations with willingness to receive the HPV vaccine, while perceived barriers (β = −0.071 and *p* < 0.001) had a negative association with willingness to receive the HPV vaccine. Having had sexual experience (β = 0.041 and *p* < 0.007) had a positive significant association with willingness to receive the HPV vaccine. The adjusted R2 was 0.094, indicating that 9.4% of willingness to receive the HPV vaccination could be explained by this model.

## 4. Discussion

This study provides the first evidence of female college students’ HPV vaccine uptake rate in mainland China. Nearly three years after the introduction of the HPV vaccines in China, the uptake rate of 11.0% (95% CI 10.1 to 11.9) among the surveyed participants implies the need to enhance HPV vaccine uptake among young women in China. Nevertheless, the rate is slightly higher than reported among young women in Hong Kong (9.7%), in 2008, when the vaccines were newly introduced to the country [23]. The HPV vaccine uptake rate has remained low in other countries in Asia, such as Singapore where only 13.6% women aged 18 to 26 years reported having been immunized against HPV [24]. In this study, based on the results of the PLS-SEM, indicators of higher socioeconomic status such as residing in the eastern region, being born in a city area, and a higher monthly disposable fund were found to correlate with the HPV vaccine uptake. This finding implies that socioeconomic status has an important role in influencing HPV vaccine uptake among female college students. There are gradually decreasing vaccination rates from eastern to western China corresponding to the decrease in socioeconomic status from the eastern to the western region. These regional differences in HPV vaccine uptake rates warrant considerable attention. A recent study showed that rural residents had higher incidence rates of cervical cancer than urban residents. Higher incidence rates occurred in central China, followed by western, and then eastern China [25]. It is important to note that although there are higher incidence rates of cervical cancer in central and western China, the HPV vaccine uptake in this study was lower among participants in the central and western regions. The HPV vaccination rates varied geographically according to the level of socioeconomic status found in this study, a finding consistent with that from Western countries [26,27]. This demonstrates the importance of scaling up HPV vaccination coverage in central and western regions, where the highest incidence rates of cervical cancer and lower rates of HPV vaccine uptake were evident.

This study also observed a greater likelihood of HPV vaccination among older than among younger female college students. This highlights the need to promote HPV vaccination among younger age college students by informing them of the importance of early protection against HPV infections. It is important to inform them of recent evidence indicating that young women in China aged 25 and below presented the highest HPV infection rate [15]. They should also be aware that protection against HPV infections by vaccination is extremely important because mounting evidence shows that sexual intercourse among young undergraduate students in China is prevalent [28,29] and that consistent condom use is reported by only slightly over half of the students with sexual experience [29].

Another important highlight of the finding was the lower proportion of correct responses for all knowledge items among those who had not taken the vaccine as compared with those who had received the HPV vaccination. The PLS-SEM also revealed a strong influence of knowledge on HPV vaccine uptake. This indicates that increasing knowledge or awareness is an important factor in raising HPV vaccination rates. Perceived susceptibility was found to have a positive association with HPV vaccine uptake, which has likewise been found in many other studies on young adults in Asia and in Western countries [25,30,31,32]. This highlights the importance of psychosocial predictors of HPV vaccine uptake in China. One possible tactic for raising the perception of susceptibility among young people is to raise their awareness of the high degree of infectiousness of HPV in China. It is of the utmost importance to inform young college women about recent data showing the high prevalence of HPV among women in China [13,14], and most importantly that young women are the group with the highest HPV infection rate [15].

The proportion of nearly 54% female college students willing to receive HPV vaccination in this study is similar to that found in a recent smaller-scale study conducted among young women in China [33]. The model of willingness to receive HPV vaccination is important to informing strategies that emphasize a socioeconomic-based approach targeting lower-income groups and rural residents. Given the evidence of low willingness to receive the HPV vaccination and high cervical cancer burden among the lower socioeconomic group [25], immediate strategies aimed at increasing HPV vaccine uptake among young women from lower socioeconomic groups in China are highly warranted. Various strategies have been suggested, and one example is the proposed “semi-mandatory HPV vaccination strategy,” which subsidizes HPV vaccination targeted at low-income settings for high-risk individuals [34]. Subsidizing the underserved population is highly recommended as a cost-effectiveness analysis evaluating the public financing of HPV vaccination found that women from lower-income quintiles gained greater financial risk protection as compared with those in the upper wealth quintiles [35]. Furthermore, regional disparities in willingness to receive the HPV vaccination found in this study demand targeted approaches to enhance the willingness to be vaccinated against HPV in high-risk geographic areas, particularly in the central and western regions of China.

The model also revealed that among all factors investigated, knowledge was identified as a strong driver of willingness to be vaccinated against HPV infection. A recent publication revealed that school-based education was effective and appropriate for increasing HPV-related knowledge and acceptability of HPV vaccines among junior middle school students in China [36]. Campus-based health education on HPV infection and cervical cancer prevention should, thus, be introduced to enrich and increase the level of knowledge and subsequent HPV vaccine coverage rate among female college students.

Our study suggests that imparting perceived susceptibility and perceived benefits could increase students’ willingness to take the HPV vaccine. Having a good understanding of the potential benefits of the HPV vaccination could outweigh perceived barriers to receiving the HPV vaccination [30]. Information regarding the benefits of the HPV vaccination should be emphasized among young people. This includes information about the prevention of HPV infection and cervical cancers as well as the reduction of the occurrence of genital warts and anal and penile cancers [32].

Our model indicated that students who have had sexual experience are more likely to express a willingness to receive the HPV vaccination. Further studies need to be conducted to find reasons for unwillingness to be vaccinated against HPV. These could include a perceived lack of risk due to not having had sexual intercourse or social stigma associated with seeking HPV vaccines, as found in other studies [37,38]. The results indicate the need to highlight the importance of HPV vaccination among female college students who have never had a sexual experience. They should be made aware that HPV vaccination is most effective when administered before exposure to any HPV infection [39].

### Limitations

This study is not without limitations. The main limitations are the nature of the cross-sectional study, which was unable to provide inferential causality, and the use of self-reported data, which can be subject to self-reporting bias. Of note, the HPV vaccine uptake rate found by this study may not be nationally representative, and therefore warrants cautious interpretation. The results should be interpreted with caution. Despite these limitations, the sample was large with diverse sociodemographic backgrounds representative of female college students from the eastern, central, and western regions of China.

## 5. Conclusions

This study found that HPV vaccination rates remain critically low among female college students approximately three years after the licensure of the HPV vaccines in China. Despite the low vaccination rate, the overall acceptability of the vaccine among the unvaccinated was moderate. Most importantly, the study found associations between geographical region, socioeconomic status, and young college women’s knowledge and beliefs about both HPV vaccine uptake and willingness to receive the HPV vaccine. The findings suggest that it is essential to establish a campus-based HPV immunization delivery program. 

## Figures and Tables

**Figure 1 vaccines-08-00031-f001:**
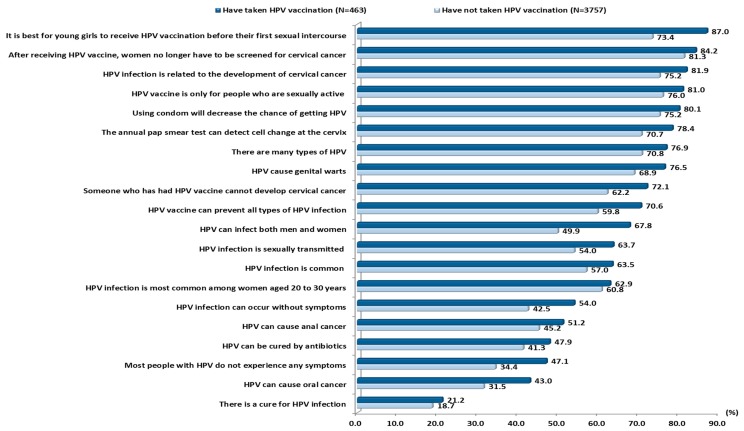
Proportion of correct responses for knowledge item between respondent who have taken HPV vaccination and who have not (*N* = 4220).

**Figure 2 vaccines-08-00031-f002:**
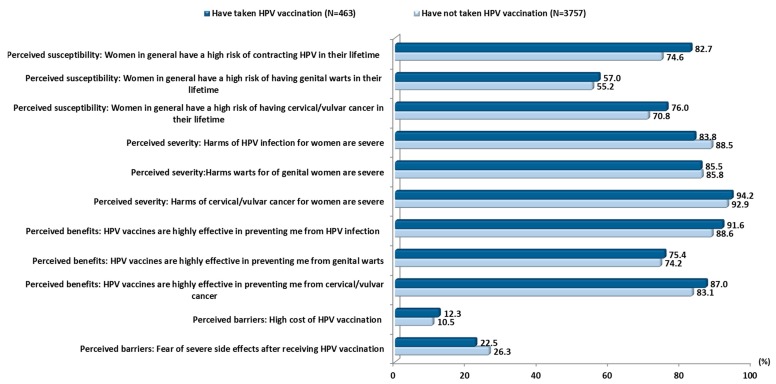
Differences in proportion of agreement to health belief statements among vaccinated and unvaccinated participants (*N* = 4220).

**Figure 3 vaccines-08-00031-f003:**
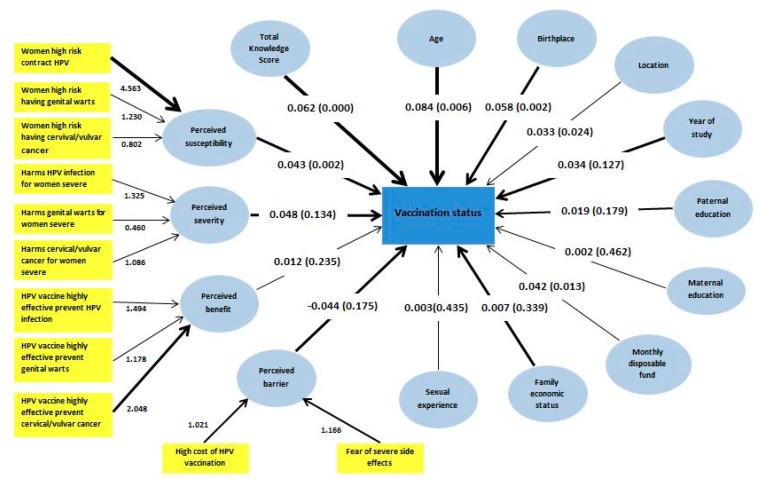
Partial least squares structural equation model of factors predicting female college students’ HPV vaccination uptake (*N* = 4220).

**Figure 4 vaccines-08-00031-f004:**
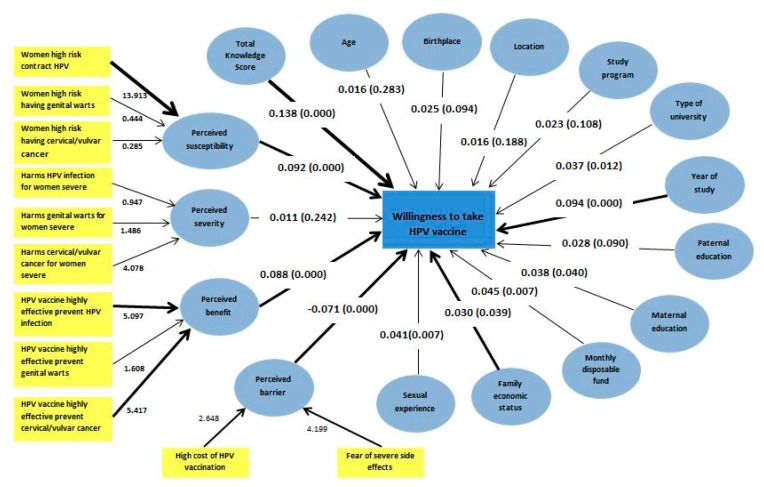
Partial least squares structural equation model of factors predicting female college students’ willingness to receive HPV (*N* = 3757).

**Table 1 vaccines-08-00031-t001:** Demographic characteristics and univariate analyses of factors associated with HPV vaccine uptake (*N* = 4220).

		Univariate Analysis		
		HPV Vaccine Uptake		
	Overall	Yes (*N* = 463)	No (*N* = 3757)	*p*-Value
**Demographic Characteristics**				
Age group (years)				
16–18	595 (14.1)	56 (9.4)	539 (90.6)	
19–22	3056 (72.4)	292 (9.6)	2764 (90.4)	*p* < 0.001
23 and above	569 (13.5)	115 (20.2)	454 (79.8)	
Ethnicity				
Han	3743 (88.7)	414 (11.1)	3329 (88.9)	0.641
Others	477 (11.3)	49 (10.3)	428 (89.7)	
Birthplace				
Village	2753 (65.2)	265 (9.6)	2488 (90.4)	
Town	757 (17.9)	78 (10.3)	679 (10.3)	*p* < 0.001
City	710 (16.8)	120 (16.9)	590 (16.9)	
Location				
East China	1622 (38.4)	222 (13.7)	1400 (86.3)	
West China	1721 (40.8)	148 (8.6)	1573 (91.4)	*p* < 0.001
Central China	877 (20.8)	93 (10.6)	784 (89.4)	
Type of university †				
Type A	2292 (54.3)	268 (11.7)	2024 (88.3)	
Type B	1272 (30.1)	130 (10.2)	1142 (89.8)	0.405
Type C	315 (7.5)	33 (10.5)	282 (89.5)	
Type D	341 (8.1)	32 (9.4)	309 (90.6)	
Year of study				
First-year	1926 (45.6)	161 (8.4)	1765 (91.6)	
Second-year	852 (20.2)	78 (9.2)	774 (90.8)	
Third-year	611 (14.5)	66 (10.8)	545 (89.2)	*p* < 0.001
Fourth-year	429 (10.2)	59 (13.8)	370 (86.2)	
Fifth-year and postgraduate	402 (9.5)	99 (24.6)	303 (75.4)	
Study program				
Medical	2954 (70.0)	322 (10.9)	2632 (89.1)	0.830
Non-medical	1266 (30.0)	141 (11.1)	1125 (88.9)	
Maternal educational level				
Elementary school and below	1439 (34.1)	146 (10.1)	1239 (89.9)	
Junior high school	1504 (35.6)	148 (9.8)	1356 (90.2)	0.019
High school or secondary school	869 (20.6)	112 (12.9)	757 (87.1)	
University and above	408 (9.7)	57 (14.0)	351 (86.0)	
Paternal educational level				
Elementary school and below	836 (19.8)	93 (11.1)	743 (88.9)	
Junior high school	1740 (41.2)	175 (10.1)	1565 (89.9)	0.001
High school or secondary school	1111 (26.3)	109 (9.8)	1002 (90.2)	
University and above	533 (12.6)	86 (16.1)	447 (83.9)	
Monthly disposable fund (RMB)				
<1000	1217 (28.8)	109 (9.0)	1108 (91.0)	
1000–1999	2326 (55.1)	230 (9.9)	2096 (90.1)	*p* < 0.001
2000 and above	677 (16.0)	124 (18.3)	553 (81.7)	
Perceived family economic status				
Rich/Intermediate	2490 (59.0)	297 (11.9)	2193 (88.1)	0.019
Poor	1730 (41.0)	166 (9.6)	1564 (90.4)	
**Sexual risk profile**				
Ever had sexual experience				
Yes	752 (17.8)	110 (14.6)	642 (85.4)	0.001
No	3468 (82.2)	353 (10.2)	3115 (89.8)	
Number of sexual partner (*n* = 752)				
1	517 (68.8)	74 (14.3)	443 (85.7)	0.739
>1	235 (31.3)	36 (15.3)	199 (84.7)	
Condom use (*n* = 752)				
Never	31 (4.1)	6 (19.4)	25 (80.6)	
Rarely	52 (6.9)	5 (9.6)	47 (90.4)	0.604
Sometimes	117 (15.6)	19 (16.2)	98 (83.8)	
Always	552 (73.4)	80 (14.5)	472 (85.5)	
History of being diagnosed with sexually transmitted infection (*n* = 752)				
Yes	69 (9.2)	12 (17.4)	57 (82.6)	0.477
No	683 (90.8)	98 (14.3)	585 (85.7)	
**Knowledge**				
Total Knowledge Score				
Score 0–12	2021 (48.1)	167 (8.2)	1861 (91.8)	*p* < 0.001
Score 13–20	2192 (51.9)	296 (13.5)	1896 (86.5)	
**Attitudes**				
**Perceived susceptibility**				
Women in general have a high risk of contracting HPV in their lifetime				
Yes	3184 (75.5)	383 (12.0)	2801 (88.0)	*p* < 0.001
No	1036 (24.5)	80 (7.7)	956 (92.3)	
Women in general have a high risk of having genital warts in their lifetime				
Yes	2338 (55.4)	264 (11.3)	2074 (88.7)	0.488
No	1882 (44.6)	199 (10.6)	1683 (89.4)	
Women in general have a high risk of having cervical/vulvar cancer in their lifetime				
Yes	3013 (71.4)	352 (11.7)	2661 (88.3)	0.019
No	1207 (28.6)	111 (9.2)	1096 (90.8)	
**Perceived severity**				
Harms of HPV infection for women are severe				
Yes	3713 (88.0)	388 (10.4)	3325 (89.6)	0.005
No	507 (12.0)	75 (14.8)	432 (85.2)	
Harms of genital warts for women are severe				
Yes	3620 (85.8)	396 (10.9)	3224 (89.1)	0.888
No	600 (14.2)	67 (11.2)	533 (88.8)	
Harms of cervical/vulvar cancer for women are severe				
Yes	3925 (93.0)	436 (11.1)	3489 (88.9)	0.335
No	295 (7.0)	27 (9.2)	268 (90.8)	
**Perceived benefit**				
HPV vaccines are highly effective in preventing me from HPV infection				
Yes	3751 (88.9)	424 (11.3)	3327 (88.7)	0.050
No	469 (11.1)	39 (8.3)	430 (91.7)	
HPV vaccines are highly effective in preventing me from genital warts				
Yes	3138 (74.4)	349 (11.1)	2789 (88.9)	0.612
No	1082 (25.6)	114 (10.5)	968 (89.5)	
HPV vaccines are highly effective in preventing me from cervical/vulvar cancer				
Yes	3525 (83.5)	403 (11.4)	3122 (88.6)	0.033
No	695 (16.5)	60 (8.6)	635 (91.4)	
**Perceived barriers**				
High cost of HPV vaccination				
Yes	451 (10.7)	57 (12.6)	394 (87.4)	0.232
No	3769 (89.3)	406 (10.8)	3363 (89.2)	
Fear of severe side effects				
Yes	1091 (25.9)	104 (9.5)	987 (90.5)	0.081
No	3129 (74.1)	359 (11.5)	2770 (88.5)	

† Types of University: Type A, National key universities and provincial key colleges; Type B, general undergraduate universities (national enrollment); Type C, local undergraduate universities (provincial enrollment); and Type D, technical universities.

**Table 2 vaccines-08-00031-t002:** Univariate analyses of factors associated with intention to take HPV vaccination (*N* = 3757).

	Frequency (%)	Univariate		
		Intention to Take HPV Vaccination		
		Yes (*n* = 2010)	No (*n* = 1747)	*p*-Value
**Demographic Characteristics**				
Age group (years old)				
16–18	539 (14.3)	269 (49.9)	270 (50.1)	
19–22	2764 (73.6)	1444 (52.2)	1320 (47.8)	*p* < 0.001
23 and above	454 (12.1)	297 (65.4)	157 (34.6)	
Ethnicity				
Han	3329 (88.6)	1773 (53.3)	1556 (46.7)	0.440
Others	428 (11.4)	237 (55.4)	191 (44.6)	
Birthplace				
Village	2488 (66.2)	1231 (49.5)	1257 (50.5)	
Town	679 (18.1)	411 (60.5)	268 (39.5)	*p* < 0.001
City	590 (15.7)	368 (62.4)	222 (37.6)	
Location of school				
East China	1400 (37.3)	796 (56.9)	604 (43.1)	
West China	1573 (41.9)	852 (54.2)	721 (45.8)	*p* < 0.001
Central China	784 (20.9)	362 (46.2)	422 (53.8)	
Type of university				
Level A	2024 (53.9)	1148 (56.7)	876 (43.3)	
Level B	1142 (30.4)	572 (50.1)	570 (49.9)	*p* < 0.001
Level C	282 (7.5)	136 (48.2)	146 (51.8)	
Technical level	309 (8.2)	154 (49.8)	155 (50.2)	
Year of study				
First-year	1765 (47.0)	876 (49.6)	889 (50.4)	
Second-year	774 (20.6)	374 (48.3)	400 (51.7)	
Third-year	545 (14.5)	303 (55.6)	242 (44.4)	*p* < 0.001
Fourth-year	370 (9.8)	239 (64.6)	131 (35.4)	
Fifth-year and postgraduate	303 (8.1)	218 (71.9)	85 (28.1)	
Study program				
Medical related major	2632 (70.1)	1142 (54.8)	1190 (45.2)	0.017
Non-medical related major	1125 (29.9)	568 (50.5)	557 (49.5)	
Maternal educational level				
Elementary school and below	1239 (34.4)	612 (47.3)	681 (52.7)	
Junior high school	1356 (36.1)	715 (52.7)	641 (47.3)	*p* < 0.001
High school or secondary school	757 (20.1)	463 (61.2)	294 (38.8)	
University and above	351 (9.3)	220 (62.7)	131 (37.3)	
Paternal educational level				
Elementary school and below	743 (19.8)	352 (47.4)	391 (52.6)	
Junior high school	1565 (41.7)	788 (50.4)	777 (49.6)	*p* < 0.001
High school or secondary school	1002 (26.7)	583 (58.2)	419 (41.8)	
University and above	447 (11.9)	287 (64.2)	160 (35.8)	
Monthly disposable fund (RMB)				
<1000	1108 (29.5)	526 (47.5)	582 (52.5)	
1000–1999	2096 (55.8)	1116 (53.2)	980 (46.8)	*p* < 0.001
2000 and above	553 (14.7)	368 (66.5)	185 (33.5)	
Perceived family economic status				
Rich/Intermediate	2193 (58.4)	1250 (57.0)	943 (43.0)	*p* < 0.001
Poor	1564 (41.6)	760 (48.6)	804 (51.4)	
**Sexual risk profile**				
Ever had sexual experience				
Yes	642 (17.1)	397 (61.8)	245 (38.2)	*p* < 0.001
No	3115 (82.9)	1613 (51.8)	1502 (48.2)	
**HPV knowledge**				
Knowledge score 0-12	1861 (49.5)	855 (45.9)	1006 (54.1)	*p* < 0.001
Knowledge score 13-20	1896 (50.5)	1155 (60.9)	741 (39.1)	
**Attitudes**				
**Perceived susceptibility**				
Women in general have a high risk of contracting HPV in their lifetime				
Yes	2801 (74.6)	1611 (57.5)	1190 (42.5)	*p* < 0.001
No	956 (25.4)	399 (41.7)	557 (58.3)	
Women in general have a high risk of having genital warts in their lifetime				
Yes	2074 (55.2)	1155 (55.7)	919 (44.3)	0.003
No	1683 (44.8)	855 (50.8)	828 (49.2)	
Women in general have a high risk of having cervical/vulvar cancer in their lifetime				
Yes	2661 (70.8)	1481 (55.7)	1180 (44.3)	*p* < 0.001
No	1096 (29.2)	529 (48.3)	567 (51.7)	
**Perceived severity**				
Harms of HPV infection for women are severe				
Yes	3325 (88.5)	1799 (54.1)	1526 (45.9)	0.040
No	432 (11.5)	211 (48.8)	221 (51.2)	
Harms of genital warts for women are severe				
Yes	3224 (85.8)	1733 (53.8)	1491 (46.2)	0.454
No	533 (14.2)	277 (52.0)	256 (48.0)	
Harms of cervical/vulvar cancer for women are severe				
Yes	3489 (92.9)	1893 (54.3)	1596 (45.7)	0.001
No	268 (7.1)	117 (43.7)	151 (56.3)	
**Perceived benefit**				
HPV vaccines are highly effective in preventing me from HPV infection				
Yes	3327 (88.6)	1850 (55.6)	1477 (44.4)	*p* < 0.001
No	430 (11.4)	160 (37.2)	270 (62.8)	
HPV vaccines are highly effective in preventing me from genital warts				
Yes	2789 (74.2)	1542 (55.3)	1247 (44.7)	*p* < 0.001
No	968 (25.8)	468 (48.3)	500 (51.7)	
HPV vaccines are highly effective in preventing me from cervical/vulvar cancer				
Yes	3122 (83.1)	1754 (56.2)	1368 (43.8)	*p* < 0.001
No	635 (16.9)	256 (40.3)	379 (59.7)	
**Perceived barriers**				
High cost of HPV vaccination				
Yes	394 (10.5)	179 (45.4)	215 (54.6)	0.001
No	3363 (89.5)	1831 (54.4)	1532 (45.6)	
Fear of severe side effects after receiving HPV vaccination				
Yes	987 (26.3)	474 (48.0)	513 (52.0)	*p* < 0.001
No	2770 (73.7)	1536 (55.5)	1234 (44.5)	

Types of University: Type A, National key universities and provincial key colleges; Type B, general undergraduate universities (national enrollment); Type C, local undergraduate universities (provincial enrollment); and Type D, technical universities.

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
