# Peer review of "Human Papillomavirus (HPV) Vaccine Uptake and the Willingness to Receive the HPV Vaccination among Female College Students in China: A Multicenter Study"

_vaccines, 2020, doi:10.3390/vaccines8010031_

Round 1
Reviewer 1 Report
The manuscript from You et al, details a public health survey of female college students across different regions of China to determine human papillomavirus (HPV) vaccine uptake and willingness to receive HPV vaccination. The data suggest that the geographical region and socio-economic differences play an important role in HPV vaccination rates and willingness to receive the HPV vaccine.
HPV and associated cervical cancer are a public health issue world-wide. This article identifies some parameters that are linked to vaccination rates and the willingness to be vaccinated. This article will be of great interest for individuals in public health roles. It was very well written and well organized with only a couple of issues.
I was not able to locate Figure 2 in the manuscript. The page number seems off. I appears that the document is renumbered after section or page breaks.
Reviewer 2 Report
This is an interesting study. However, some points should be revised.
1. The actual vaccination uptake in the population is uncertain because this study is a questionnaire-based survey. Is HPV vaccination uptake (11%) obtained from this study similar to the vaccination rate in authors’ country? If so, many data investigated in this study will be highly reliable. If not so, these data may not be reliable.
2. The reviewer recommends that P-values, but not b-values, are presented in Abstract.
3. Figure 2 was not presented in the text.
4. The reference number was not presented in Reference section.
Round 2
Reviewer 2 Report
The authors revised the manuscript according to the reviewer's comments.